# Overcoming stability challenges during continuous intravenous administration of high-dose amoxicillin using portable elastomeric pumps

**Guillaume Binson** [1,2⊗], **Claire Grignon** [1⊗], **Gwenaël Le Moal** [3], **Pauline Lazaro** [1], **Jérémy Lelong** [4], **France Roblot** [3], **Nicolas Venisse** [2,4], **Antoine Dupuis** [1,2]*

**1** Department of Pharmacy, University Hospital of Poitiers, Poitiers, France, **2** CIC Inserm, Poitiers, France, **3** Department of Infectious Diseases, University Hospital of Poitiers, Poitiers, France, **4** Department of Pharmacokinetics, University Hospital of Poitiers, Poitiers, France

⊗ These authors contributed equally to this work.

* antoine.dupuis@univ-poitiers.fr

**Data Availability Statement:** All relevant data are within the manuscript.

## Abstract

While treatment of serious infectious diseases may require high-dose amoxicillin, continuous infusion may be limited by lack of knowledge regarding the chemical stability of the drug. Therefore, we have performed a comprehensive study so as to determine the chemical stability of high-dose amoxicillin solutions conducive to safe and effective continuous intravenous administration using portable elastomeric pumps. First, amoxicillin solubility in water was assessed within the range of 25 to 300 mg/mL. Then, amoxicillin solutions were prepared at different concentrations (25, 50, 125, 250 mg/mL) and stored in different conditions (5±2˚C, 25±1˚C, 30±1˚C and 37±1˚C) to investigate the influence of concentration and temperature on the chemical stability of amoxicillin. Finally, its stability was assessed under optimized conditions using a fully validated HPLC-UV stability-indicating method. Degradation products of amoxicillin were investigated by accurate mass determination using high-resolution mass spectrometry. Amoxicillin displayed limited water solubility requiring reconstitution at concentrations below or equal to 150 mg/mL. Amoxicillin degradation were time, temperature as well as concentration-dependent, resulting in short-term stability, in particular at high concentrations. Four degradation products of amoxicillin have been identified. Among them, amoxicilloic acid and diketopiperazine amoxicillin are at risk of allergic reaction and may accumulate in the patient. Optimized conditions allowing for continuous infusion of high-dose amoxicillin has been determined: amoxicillin should be reconstituted at 25 mg/mL and stored up to 12 hours at room temperature (22 ± 4˚C) or up to 24 hours between 4 and 8˚C.

## Introduction

The management of several serious infectious diseases such as bone and joint infections as well as infective endocarditis, requires intravenous administration of high-dose amoxicillin (100–300 mg/kg/day) over a prolonged period, ranging from a few weeks to several months [1–4].

**Funding:** The authors received no specific funding for this work.

**Competing interests:** The authors have declared that no competing interests exist.

Amoxicillin is a β-lactam antibiotic. β-Lactams are time-dependent antibiotics, meaning that their efficacy depends on the time that free serum concentrations remain above the minimal inhibitory concentration (MIC) during the dosing interval [5]. It has been demonstrated that continuous infusion maintains concentrations above the MIC for a longer period of time within the dosing interval [6]. Moreover, mounting evidence from clinical studies indicates that continuous infusion of time-dependent β-lactam antibiotics may improve clinical success [7–9]. Therefore it may be well-founded to administrate β-Lactams using continuous infusion in patients suffering from serious infectious disease. However, despite the possible clinical benefits of this mode of administration, one of the practical concerns related to continuous infusion is the limited stability of certain antibiotic agents. Indeed, stability issues have to be taken into consideration when implementing drug administration, in order to ensure drug efficacy and safety. Regarding β-lactam antibiotics, not knowing how long they remain stable during infusion may be a limiting factor for continuous administration [10,11].

Furthermore, for several decades, intravenous antimicrobials have been administered increasingly in outpatient settings, in particular thanks to the use of portable devices [12,13]. Outpatient parenteral antimicrobial therapy (OPAT) allows for early hospital discharge, and further reduces costs with fewer nursing and clinic visits [14]. Moreover, OPAT improves quality of life, and portable elastomeric pumps gives patients more flexibility and control over their treatment [15]. Among the important aspects described in OPAT practice guidelines, drug stability has been underlined as a crucial point to be taken into consideration to ensure efficacy and safety of antimicrobial therapy [12,16,17]. In a recent survey, osteomyelitis, prosthetic joint infections and endocarditis were the most commonly reported indications for OPAT [18]. In treatment of these infectious diseases, continuous infusion of high-dose amoxicillin may be limited by lack of knowledge of the chemical stability of the drug. Indeed, very few studies are available in the literature regarding the stability of amoxicillin in aqueous solution for intravenous administration and results regarding long-term stability have been inconsistent [19–23].

The aim of this work was to propose safe and effective conditions for continuous intravenous administration of high-dose amoxicillin using portable elastomeric pumps. For that purpose we performed a comprehensive study designed to determine the chemical stability of high-dose amoxicillin solutions.

## Materials and methods

### Chemicals and infusion devices

Amoxicillin powder used for calibration of the method was purchased from Sigma-Aldrich (Sigma-Aldrich, France) while amoxicillin sodium powder for solution for injection, equivalent to amoxicillin 2 g, was used for the pharmaceutical preparation (Panpharma, France). HPLC-grade methanol was obtained from Carlo Erba (Carlo Erba, France) and ultrapure water was provided using a Millipore Direct-Q 3 UV water purification system (MerckMillipore, France). Sterile water and 0.9% sodium chloride for injection were obtained from B. Braun (B.Braun, France).

Portable elastomeric pumps Infusor (48 mL, 2 mL/h) and FOLFusor (240 mL, 10 mL/h), were obtained from Baxter (Baxter, France) and portable elastomeric pumps Accufuser (480 mL, 20 mL/h) from Wym (Wym, France). The Infusor and FOLFusor reservoirs are made of synthetic polyisoprene and the Accufuser reservoir is made of medical silicone.

### Solubility study

Amoxicillin solubility was assessed within the range 25 to 300 mg/mL by dissolving a vial of amoxicillin sodium, equivalent to amoxicillin 2 g, in adequate volume of sterile water for

injection. The vial was vortexed for 10 min, centrifugated at 3500 G for 10 min and the amoxicillin concentration was determined in the supernatant. Each experiment was conducted in triplicate.

## Stability study

To determine the optimal volume of dilution for amoxicillin reconstitution, we investigated the influence of the concentration on the chemical stability of amoxicillin. Amoxicillin solutions were prepared at different concentrations in various infusion devices (Table 1). The filled elastomeric pumps were stored at 25 ± 1˚C for 24 hours in a climate chamber without humidity control (Air concept, FirLabo, France). Samples (n = 3) were collected at different times over the 24-hour storage period and determination of the amoxicillin concentration was performed immediately. Each experiment was conducted in triplicate.

To determine the best storage conditions during amoxicillin infusion we investigated the influence of temperature on the chemical stability of amoxicillin. Amoxicillin solutions were prepared at a concentration of 125 mg/mL (6 g of amoxicillin were reconstituted with 48 ml of sterile water for injection), in order to fill an Infusor (nominal volume of 48 mL, flow rate of 2 mL/h). The filled elastomeric pumps were then stored for 24 hours at different temperature conditions: 5 ± 2˚C in a refrigerated chamber (Precision, Thermo Scientific, France), 25 ± 1˚C, 30 ± 1˚C and 37 ± 1˚C, in a climate chamber (Air concept, FirLabo, France). Samples (n = 3) were collected at different times over the 24-hour storage period and determination of the amoxicillin concentration was performed immediately. Each experiment was conducted in triplicate.

Finally, the stability of amoxicillin was assessed under optimized conditions. For this purpose, amoxicillin solutions were prepared at a concentration of 25 mg/mL (12 g of amoxicillin were reconstituted using 240 ml of sterile water for injection and 240 mL of 0.9% sodium chloride for injection), in order to fill elastomeric pumps (Accufuser, nominal volume of 480 mL, flow rate of 20 mL/h). The filled elastomeric pumps were then stored for 24 hours at room temperature (22 ± 4˚C) or between 4 and 8˚C in a refrigerated bag. Samples (n = 3) were collected at different times over the storage period and determination of the amoxicillin concentration was performed immediately. Each experiment was conducted in triplicate.

The degradation rate was estimated using the slope of the linear regression curve corresponding to amoxicillin remaining (% of initial concentration) versus time profile.

Stability was defined as less than 10% disappearance of the amoxicillin concentration, in compliance with the provisions of the US Pharmacopoeia concerning the acceptable limit of content of drug preparation settled at 90% [24], and with the European Pharmacopoeia, requiring that β-Lactams solutions always contain at least 90% of intact molecule [25].

## Amoxicillin determination

All samples were diluted to a concentration of 100 μg/mL with ultrapure water and assayed for amoxicillin concentration using a high-performance liquid chromatography method coupled

**Table 1. Amoxicillin solutions prepared to investigate the influence of concentration on the chemical stability of amoxicillin.**

| Amoxicillin concentration (mg/mL) | Amoxicillin (g) | Solvent | Solvent volume (mL) | Infusion device: nominal volume, flow rate |
|---|---|---|---|---|
| 250 | 12 | Sterile water for injection | 48 | Infusor: 48 mL, 2 mL/h |
| 125 | 6 | Sterile water for injection | 48 | Infusor: 48 mL, 2 mL/h |
| 50 | 12 | Sterile water for injection | 240 | FOLfusor: 240 mL, 10mL/h |
| 25 | 6 | Sterile water for injection : 0.9% sodium chloride (50:50, v:v) | 240 | FOLfusor: 240 mL, 10mL/h |

to UV detection (HPLC-UV). The Elite LaChrom system (VWR, France) included a binary pump (Primaid 1110) used in isocratic mode, a single wavelength ultraviolet detector (L-2400), and an autosampler (L-2200) and was controlled using EZChrom Elite 3.3 software. Separation was performed using a Nucleosil C8 analytical column (5 μm, 150 x 4.6 mm, VWR, France). Mobile phase consisted of 20% methanol and 80% ultrapure water; flow rate was set at 1 mL/min and 10 μL of the diluted sample were injected onto the column. Quantification was performed by integration of the peak at a detection wavelength of 225 nm. The stability-indicating HPLC-UV method was validated in accordance with the guidelines of the International Conference on Harmonisation of Technical Requirements for Registration of Pharmaceuticals for Human Use Q2(R1) [26]. Briefly, the method was linear over the range 0–200 μg/mL ($r^2 > 0.9993$) and the limit of quantification of amoxicillin was equal to 12.5 μg/mL. Precision of the method was high, according to intra-day and inter-day coefficients of variation, calculated at low and high concentrations, equal to or less than 4.4% and to the trueness, assessed using the bias, ranging from 88 to 108%. To ensure that the method could be regarded as suitably stability-indicating, we checked to be sure that the decomposition products obtained from amoxicillin solution subjected to severe stress (90°C, pH 1, pH10) did not coelute with the intact drug.

### Degradation product identification

Degradation products of amoxicillin were investigated by accurate mass determination using high-resolution mass spectrometry (HRMS). Briefly, amoxicillin solutions (50 mg/mL, extemporaneous and kept stored 48 hours at 37°C) were injected onto the HPLC system connected to an ultra-high definition quadrupole time-of-flight mass spectrometer (Xevo QTof, Waters, France). The mass spectrometer was equipped with electrospray source, operating in positive ion mode, using the following operating parameters: capillary voltage: 0.5 kV; sample cone voltage: 20 V; source temperature: 120°C; desolvation temperature: 600°C; cone gas flow: 50L/h; desolvatation gas flow: 1000L/h. Accurate mass measurements of main ions (Low energy: 4 eV) and fragments (high-energy ramp: 10 to 40 eV) were used to identify the major degradation products. LC-MS-measured accurate mass spectra were recorded across the range 50–1000 m/z with scan time settled at 0.1 s. The degradation products were identified by structure elucidation using the molecular ion exact mass determination and the collision-induced dissociation fragments obtained using HRMS.

## Results

### Solubility study

From 0 to 150 mg/mL, measured concentrations of amoxicillin are well-correlated to theoretical concentrations (Fig 1). At concentrations greater than 150 mg/mL, measured amoxicillin concentrations are systematically lower than theoretical values, underlining the incomplete solubility of amoxicillin in water at high concentration.

### Stability study

Whatever the nominal concentration prepared and whatever the storage conditions selected, amoxicillin concentration decreases over time, demonstrating that amoxicillin is unstable in aqueous solution.

Amoxicillin degradation rate increases depending on the initial concentration: from -0.75%/h to -3.34%/h at 25 mg/mL and 250 mg/mL respectively, leading to concentrations after 24-hour storage ranging from 83% to 13% of the initial concentration (Fig 2). Amoxicillin

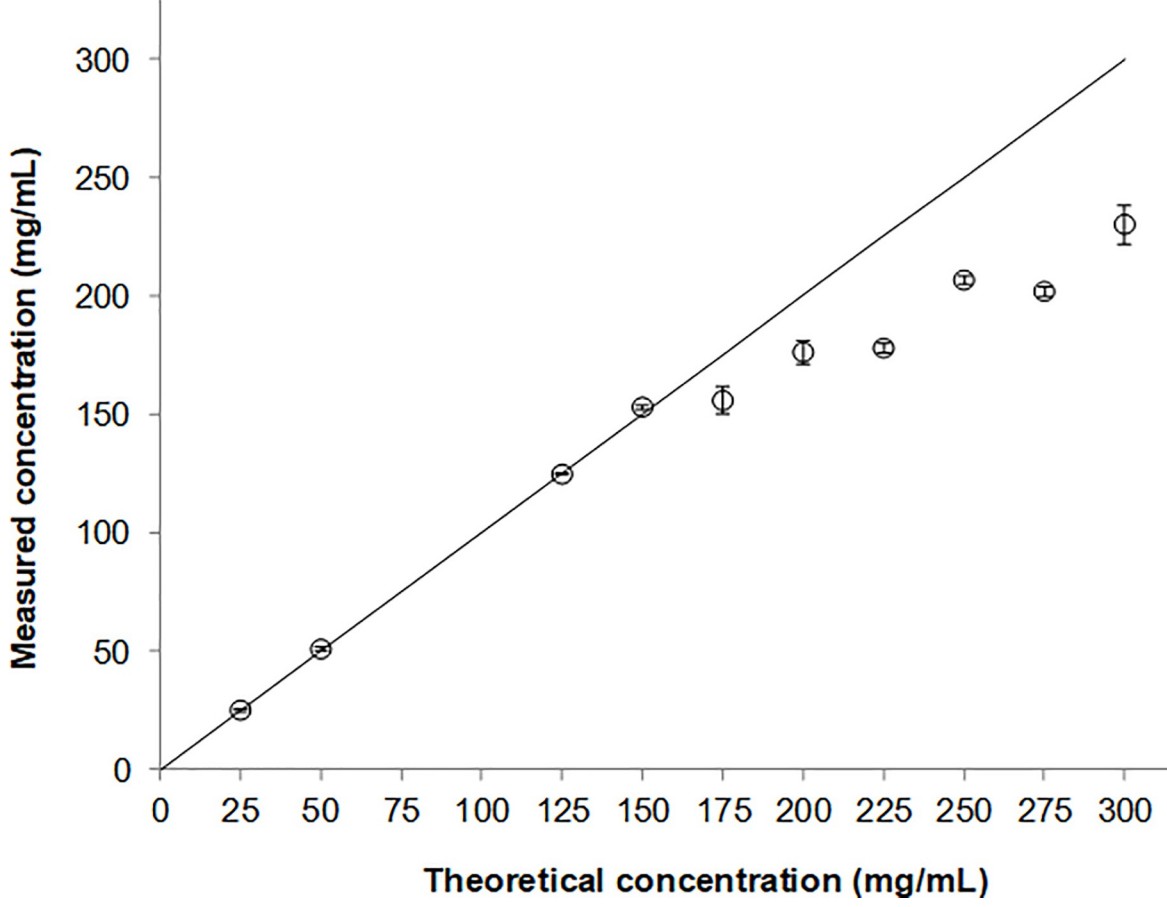

**Fig 1. Solubility of amoxicillin expressed as measured amoxicillin concentrations (○) and expected amoxicillin concentrations (solid line) versus theoretical concentration.** Each value is the mean of three independent determinations ± standard deviation.

chemical stability depends on initial concentration: the higher the concentration, the less stable the antibiotic.

Moreover, amoxicillin degradation rate increases depending on storage temperature: from -1.92%/h to -3.29%/h at 5°C and 37°C mg/mL respectively, leading to concentrations after 24-hour storage ranging from 50% to 16% of the initial concentration (Fig 3). Regarding temperature amoxicillin chemical stability is temperature-dependent: the higher the temperature, the less stable the antibiotic.

Using optimized conditions of administration, the amoxicillin remaining was greater than 90% up to 12 hours of storage at room temperature (22 ± 4°C) and up to 24 hours stored between 4 and 8°C (Table 2).

## Degradation product identification

Four degradation products of amoxicillin were identified using HRMS according to their exact mass and pattern of fragmentation (precursor → fragment ions *m/z*): amoxicillin penilloic acid (340.1331→323.1059/305.0952/189.0691), diketopiperazine amoxicillin (366.1124→160.0426/114.0374), Amoxicillin penicilloic acid (amoxicilloic acid) (384.1223→367.0954/323.1056/305.09520), and hydroxyphenylglycyl amoxicillin (515.1588→498.1361/349.0988/339.0988) (Figs 4 and 5).

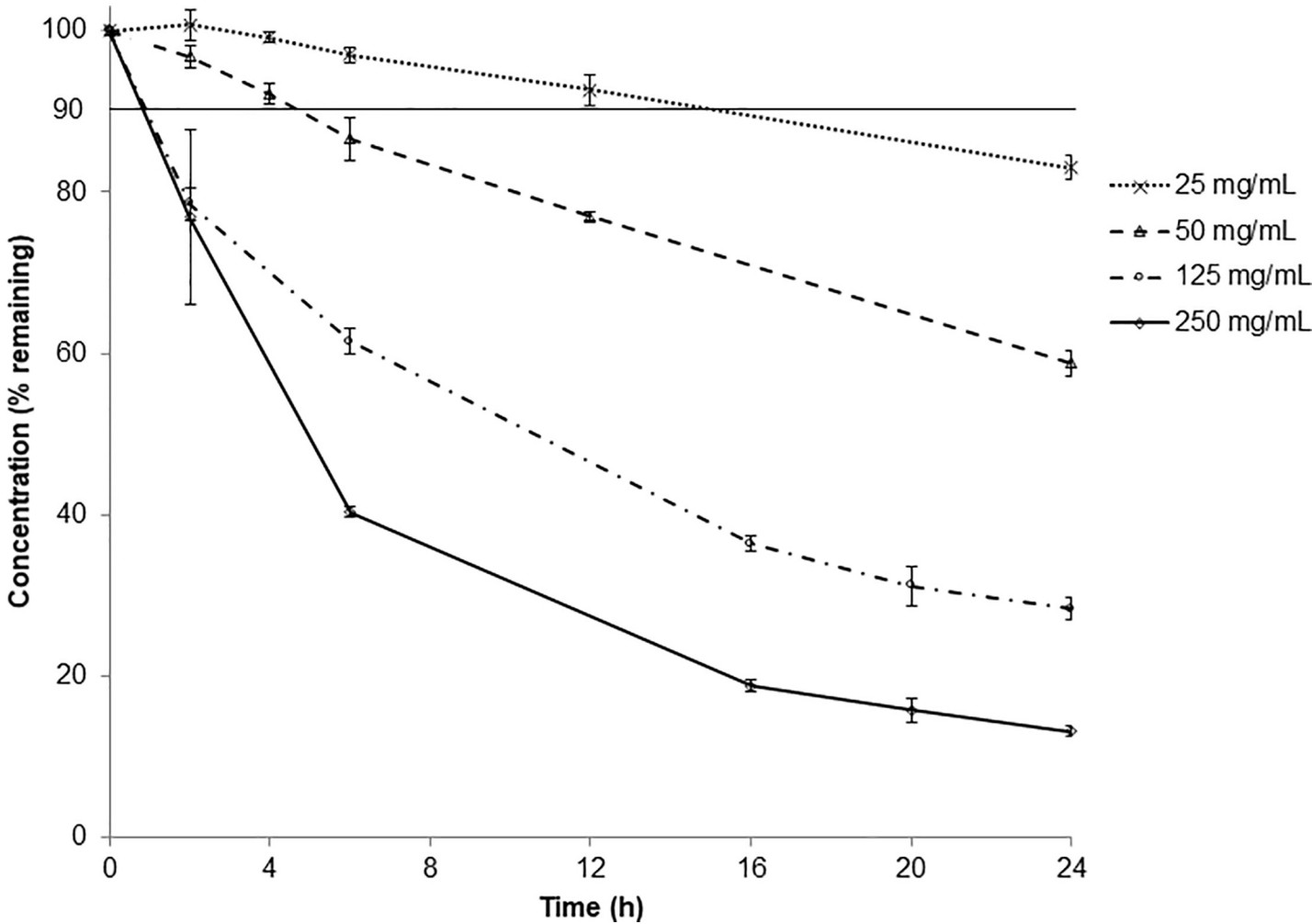

**Fig 2. Chemical stability of amoxicillin prepared at different concentrations in portable elastomeric pump stored at 25 ± 1°C.** Values are expressed as mean ± standard deviation. The horizontal line indicates the limit set by the Pharmacopoeias (90% of initial drug concentration).

## Discussion

The high doses of amoxicillin required for treatment of severe infections should be administrated over a long period, and preferably by continuous intravenous administration [1–9]. According to the limited volume of medical devices available for continuous drug infusion, particularly the portable devices used for OPAT, highly concentrated solutions of amoxicillin need to be prepared, leading to issues regarding solubility. In this study, we have demonstrated for the first time that amoxicillin has to be reconstituted at concentrations below or equal to 150 mg/mL to achieve complete solubility, which is mandatory in order to ensure accurate drug dosage and efficacy when using intravenous administration mode. Moreover, incomplete solubility of a drug may lead to particles in solutions, which may induce serious harmful effects such as pulmonary embolism or systemic inflammatory response syndrome [27–29]. Therefore, an appropriate volume of solvent is required for the reconstitution of amoxicillin to ensure safe administration, excluding the use of low-volume devices such as syringes or small volume portable pumps, when high doses of amoxicillin are to be administrated.

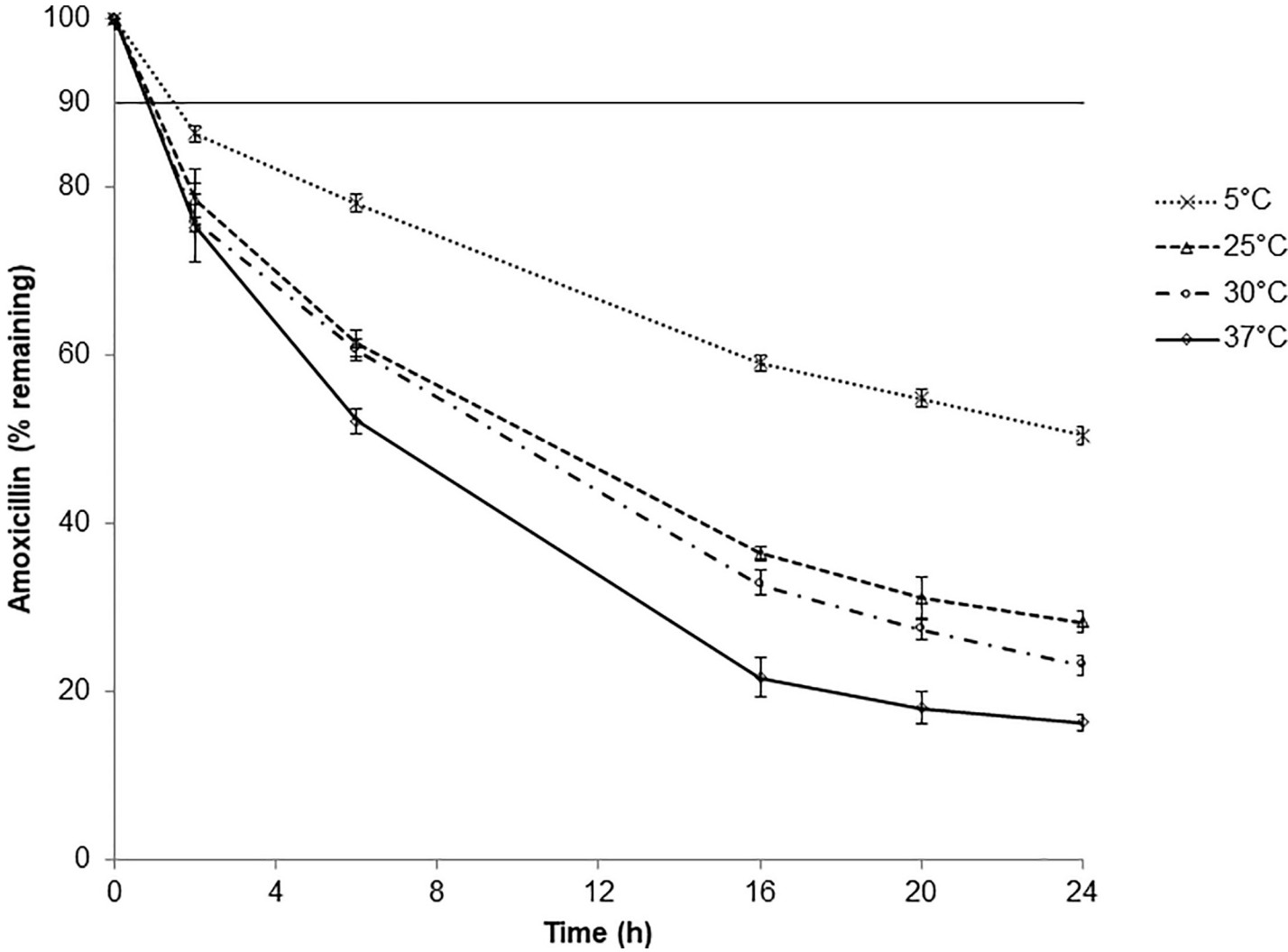

**Fig 3. Chemical stability of amoxicillin (125 mg/mL) in portable elastomeric pump stored at different temperatures.** Values are expressed as mean ± standard deviation. The horizontal line indicates the limit set by the Pharmacopoeias (90% of initial drug concentration).

This work confirms that amoxicillin degradation is time, temperature and concentration-dependent, resulting in short-term stability of amoxicillin solutions for infusion, particularly at high concentrations. Consequently, continuous infusion of the high-dose amoxicillin recommended to treat serious infectious diseases requires applying rigorous conditions of administration especially for OPAT. Elastomeric pumps are commonly used in the setting of OPAT and are available in different volumes and flow rates to allow proper administration of a drug [30]. Therefore, we have investigated different volumes and flow rates through the use of different devices in order to determine which one best helps to achieve amoxicillin concentration-dependent stability. Regarding storage temperature, elastomeric pumps may be used in different conditions, refrigerated or not. Moreover, it has been demonstrated that the temperature of drug solutions in elastomeric pumps may dramatically increase over the day, especially for continuous infusion, reaching a maximum near body temperature [31]. In this context, we have investigated the impact of potentially different storage temperatures on amoxicillin stability in order to determine the optimal conditions for its administration.

**Table 2. Chemical stability of amoxicillin (25 mg/mL) in portable elastomeric pump stored at room temperature (22 ± 4˚C) and under refrigerated conditions (4–8˚C).** Values are expressed as mean ± standard deviation.

| Hours | Storage conditions | |
|---|---|---|
| | Room temperature | Refrigerated |
| 0 (mg/mL) | 25.65 ± 1.98 | 25.01 ± 0.58 |
| 2 (% remaining) | 100.78 ± 1.94 | 97.12 ± 4.32 |
| 4 (% remaining) | 99.17 ± 0.66 | 95.92 ± 4.37 |
| 6 (% remaining) | 97.01 ± 0.93 | 95.19 ± 4.29 |
| 12 (% remaining) | 92.63 ± 1.89 | 93.81 ± 3.21 |
| 24 (% remaining) | 83.06 ± 1.43 | 91.24 ± 1.22 |

Few previous studies have investigated amoxicillin stability in solutions, using varying solvents and concentrations, and different temperatures and containers for storage [19–23]. Most of them report limited stability unsuitable for continuous infusion over a long period. One study has assessed amoxicillin stability in elastomeric pumps and reports results contradictory to ours concerning amoxicillin degradation [21]. Indeed, the authors claim a stability of amoxicillin for 24h-48h up to 40 mg/ml, when stored at 20˚C or 35˚C. However, these conclusions do not comply with international guidelines regarding the acceptable drug degradation limit

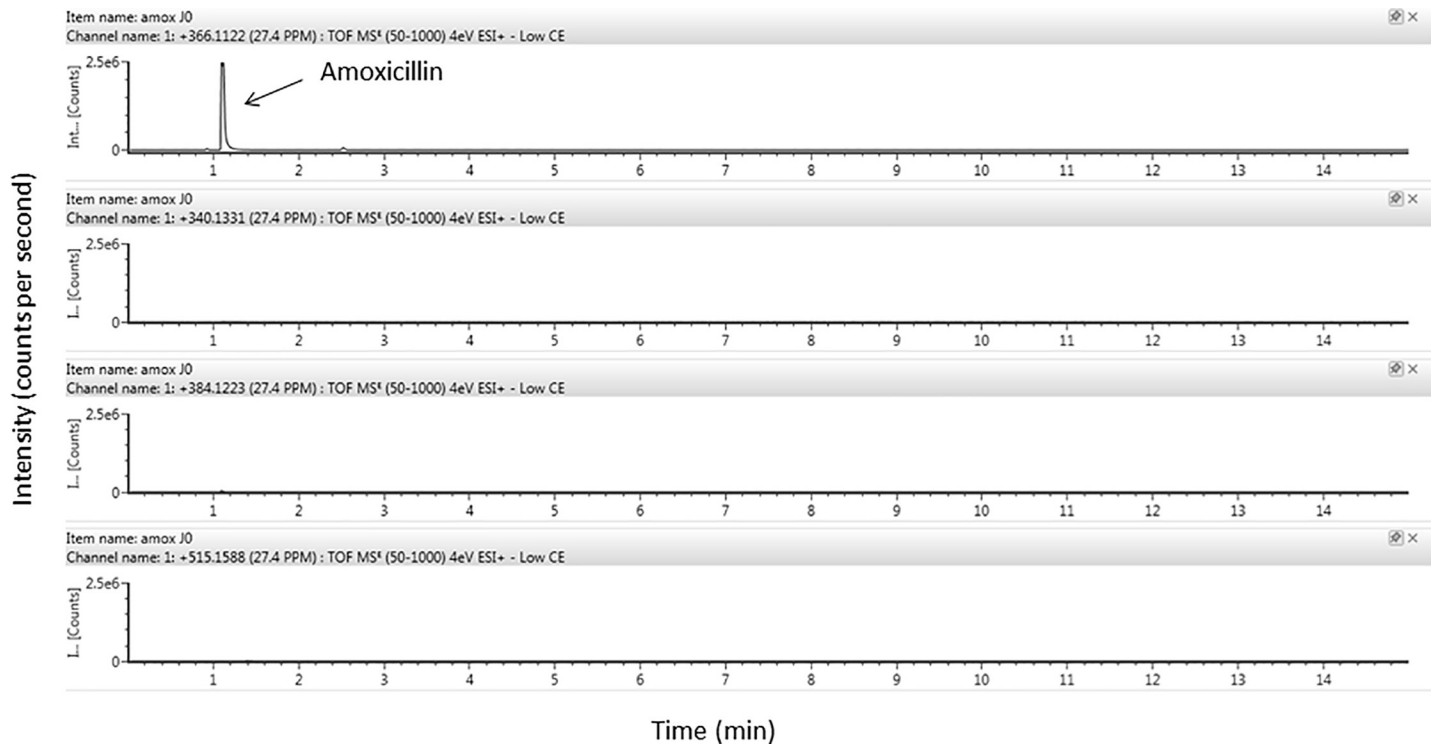

**Fig 4. Identification of amoxicillin and its degradation products using ion Trap MS/MS analysis of amoxicillin solutions (50 mg/mL) after extemporaneous preparation.**

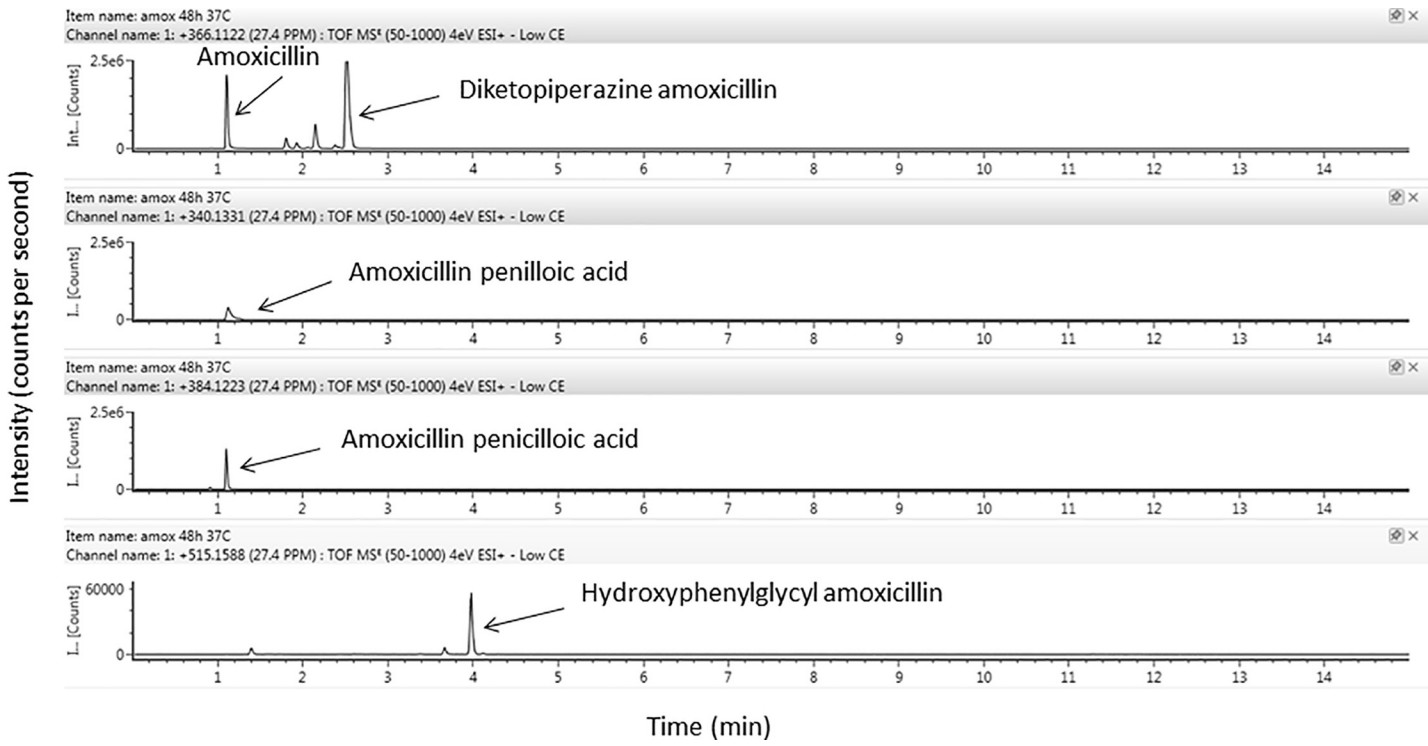

**Fig 5. Identification of amoxicillin and its degradation products using ion Trap MS/MS analysis of amoxicillin solutions (50 mg/mL) after 48-hours storage at 37˚C.**

of less than 10% [24,25]. Furthermore, even under the same conditions, the degradation rates of amoxicillin reported in this paper are lower than ours, as well as those reported by others under similar conditions [20,22,23]. In fact, it should be noted that the analytical method proposed in this study was not stability-indicating since they did not use a separation method, leading to potential confusion between amoxicillin and its degradation products [32]. This major flaw in the method is likely to explain the underestimated amoxicillin degradation rates reported by these authors.

Unreliable data regarding stability of such antibiotics may have serious consequences since clinicians may use these results to promote continuous infusion over a prolonged period. This issue raises the need for clinicians to seek pharmaceutical expertise in order to assess the level of proof of the analytical methods used to provide data having clinical impact [33].

Despite the lack of consistent data on amoxicillin stability, some authors have investigated the suitability of continuous intravenous administration of high-dose amoxicillin in patients [23,34]. However, while Verdier et al. demonstrated that continuous infusion of amoxicillin provides fewer low concentrations than intermittent administration, they reported highly inconstant amoxicillin plasma concentrations using this administration mode, despite adapted doses [34]. Moreover, despite continuous administration of high-doses amoxicillin (12 g/day), Arensdorff et al. observed, very low amoxicillin plasma concentrations (0.9 and 2.4 mg/mL) in two out of the nine patients enrolled [23]. In both these studies, amoxicillin unstability may have resulted in inaccurate dose administration, leading to suboptimal or ineffective concentrations in treated patients, with respect to the MIC of the bacteria involved in the infection. Clearly, while the great variability in amoxicillin plasma levels is due to many factors, unstability issue should not be ruled out as a causal factor.

Last but not least, drug degradation may lead to toxic byproducts [17]. Regarding amoxicillin we have identified four degradation products, Among them, amoxicilloic acid and diketopiperazine amoxicillin have been identified as the two major degradation products of amoxicillin. Hydroxyphenylglycyl amoxicillin acid and amoxicillin penilloic acid have also been described by others [35,36]. None of these degradation products have antimicrobial activity, but amoxicilloic acid and diketopiperazine amoxicillin are at risk of allergic reaction, of the same order of magnitude as those observed with amoxicillin [37,38]. In addition, it has been demonstrated that the amoxicilloic acid concentrations measured in pig kidney after intravenous administration are higher than amoxicillin concentrations and decrease more slowly, meaning that this degradation product may accumulate in the patient [39,40]. This issue could partly explain the side effects reported in patients treated with high-dose amoxicillin.

In conclusion, according to the experiments carried out in this work, the optimal conditions for the administration of continuous infusion of high-dose amoxicillin using portable elastomeric pumps have been defined. First of all, amoxicillin has to be diluted in order to obtain a final concentration of 25 mg/ml, allowing for complete dissolution and long-term stability. Dilution should be performed in sterile water for injection combined with 0.9% sodium choride for injection (50:50, v:v) in order to maintain osmolarity at a level (approximately 280 mOsm/L, data no shown) suitable for intravenous infusion using peripheral access [16]. Dilution in dextrose should be avoided since lower stability using this solvent has been demonstrated [20]. Then, under these optimized conditions, the elastomeric pump may be stored at ambient temperature (22 ± 4˚C) for 12 hours, requiring filling an elastomeric pump every 12 hours for continuous infusion. In addition, it is important to store the elastomeric pump at a temperature below 26˚C, which implies not holding the device near the body or under clothing during the day, and keeping it beside the head, outside of the blankets when the patient is bedridden. Finally, amoxicillin may remain stable over the entire day under the same conditions, provided that the elastomeric pump is stored between +4˚C and +8˚C throughout the administration, using for instance a refrigerated bag; however this can lead to potential intolerance due to infusion of cold solution.

## Acknowledgments

We wish to thank Jeffrey Arsham, an American medical translator, for his highly helpful reading of our original text.

## Author Contributions

**Conceptualization:** Gwenaël Le Moal, France Roblot, Antoine Dupuis.

**Investigation:** Guillaume Binson, Claire Grignon, Jérémy Lelong.

**Methodology:** Guillaume Binson, Claire Grignon, Pauline Lazaro, Nicolas Venisse, Antoine Dupuis.

**Project administration:** Antoine Dupuis.

**Supervision:** Antoine Dupuis.

**Validation:** Nicolas Venisse.

**Writing – original draft:** Guillaume Binson, Claire Grignon.

**Writing – review & editing:** Gwenaël Le Moal, Nicolas Venisse, Antoine Dupuis.

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
