## [Decision Letter · Decision Letter 0]

5 Aug 2019

PONE-D-19-17342

Overcoming stability challenges during continuous intravenous administration of high-dose amoxicillin using portable elastomeric pumps

PLOS ONE

Dear Dr DUPUIS,

Thank you for submitting your manuscript to PLOS ONE. After careful consideration, we feel that it has merit but does not fully meet PLOS ONE’s publication criteria as it currently stands. Therefore, we invite you to submit a revised version of the manuscript that addresses the points raised during the review process.

We would appreciate receiving your revised manuscript by Sep 19 2019 11:59PM. To enhance the reproducibility of your results, we recommend that if applicable you deposit your laboratory protocols in protocols.io, where a protocol can be assigned its own identifier (DOI) such that it can be cited independently in the future. For instructions see: http://journals.plos.org/plosone/s/submission-guidelines#loc-laboratory-protocols

We look forward to receiving your revised manuscript.

Kind regards,

José das Neves

Academic Editor

PLOS ONE

Journal Requirements:

Reviewers' comments:

Reviewer's Responses to Questions

3. Have the authors made all data underlying the findings in their manuscript fully available?

Reviewer #1: No

Reviewer #2: Yes

5. Review Comments to the Author

Reviewer #1: Excellent manuscript. The methodology follows the recommendations for drug stability testing of the British Society of Antimicrobial Chemotherapy. The discussion includes a complete review of the existing literature and addresses all the important issues around the problem of drug stability in elastomeric pumps.

Reviewer #2: This paper describes the results from an experimental research work that aim to study the stability of amoxicillin intravenous preparations in elastomeric pumps. The study was well-designed and the results are presented clearly and are adequately discussed. This work is of interest for clinical practice.

Some minor improvements can be considered:

1 - in the methods section, regarding the conditions for stability testing on Climate Chambers (line 110) it should be specified if the humidity was also controlled.

2 - in the conclusion section (line 316) the information "elastomeric pump may be stored at ambient temperature for 12 hours" should be completed with the specific range of temperature related to the conclusion

3 - the last sentence of the conclusion (lines 320-322) is not clear: it seems to suggest that the elastomeric pump is kept refrigerated during administration. Although these pumps may be refrigerated for storage, drug administration is generally preceded by warming the system at room temperature.

Also, according to the presented stability data if the elastomeric pump was kept refrigerated for 24h it would not be expected to be stable for an additional period of 12h at room temperature (corresponding to a loss of drug of approximately 17% of drug). This should be clearly discussed.

---

## [Author Response · Author response to Decision Letter 0]

6 Aug 2019

We sincerely appreciate the comments of the reviewers and revised the paper according to the suggestions. Please find below the details point-by-point of the revisions in the manuscript and our responses.

Reviewer #1

“Excellent manuscript. The methodology follows the recommendations for drug stability testing of the British Society of Antimicrobial Chemotherapy. The discussion includes a complete review of the existing literature and addresses all the important issues around the problem of drug stability in elastomeric pumps.”

We sincerely appreciate the comments of the reviewer.

Reviewer #2

“This paper describes the results from an experimental research work that aim to study the stability of amoxicillin intravenous preparations in elastomeric pumps. The study was well-designed and the results are presented clearly and are adequately discussed. This work is of interest for clinical practice. Some minor improvements can be considered:»

Thank you very much for your valuable and constructive suggestions. We have carefully revised the manuscript according to the suggestions

“1 - in the methods section, regarding the conditions for stability testing on Climate Chambers (line 110) it should be specified if the humidity was also controlled.”

According to reviewer’s suggestion, the sentence has been modified as follow: “The filled elastomeric pumps were stored at 25 ± 1°C for 24 hours in a climate chamber without humidity control (Air concept®, FirLabo, France).”

“2 - in the conclusion section (line 316) the information "elastomeric pump may be stored at ambient temperature for 12 hours" should be completed with the specific range of temperature related to the conclusion.”

According to reviewer’s suggestion, the range of temperature has been added: “Then, under these optimized conditions, the elastomeric pump may be stored at ambient temperature (22 ± 4°C) for 12 hours…”

“3 - the last sentence of the conclusion (lines 320-322) is not clear: it seems to suggest that the elastomeric pump is kept refrigerated during administration. Although these pumps may be refrigerated for storage, drug administration is generally preceded by warming the system at room temperature. Also, according to the presented stability data if the elastomeric pump was kept refrigerated for 24h it would not be expected to be stable for an additional period of 12h at room temperature (corresponding to a loss of drug of approximately 17% of drug). This should be clearly discussed.”

The sentence has been modified in order to clarify this point : “Finally, amoxicillin may remain stable over the entire day under the same conditions, provided that the elastomeric pump is stored between +4°C and +8°C throughout the administration, using for instance a refrigerated bag ; however this can lead to potential intolerance due to infusion of cold solution.

---

## [Editor Report · Decision Letter 1]

7 Aug 2019

Overcoming stability challenges during continuous intravenous administration of high-dose amoxicillin using portable elastomeric pumps

PONE-D-19-17342R1

Dear Dr. DUPUIS,

We are pleased to inform you that your manuscript has been judged scientifically suitable for publication and will be formally accepted for publication once it complies with all outstanding technical requirements.

With kind regards,

José das Neves

Academic Editor

PLOS ONE
---

## [Editor Report · Acceptance letter]

9 Aug 2019

PONE-D-19-17342R1 

Overcoming stability challenges during continuous intravenous administration of high-dose amoxicillin using portable elastomeric pumps 

Dear Dr. Dupuis:

I am pleased to inform you that your manuscript has been deemed suitable for publication in PLOS ONE. Congratulations! Your manuscript is now with our production department. 

With kind regards,

on behalf of

Dr. José das Neves 

Academic Editor

PLOS ONE